# Unbounded cache model for online language modeling with open vocabulary

**Edouard Grave**
Facebook AI Research
egrave@fb.com

**Moustapha Cisse**
Facebook AI Research
moustaphacisse@fb.com

**Armand Joulin**
Facebook AI Research
ajoulin@fb.com

## Abstract

Recently, continuous cache models were proposed as extensions to recurrent neural network language models, to adapt their predictions to local changes in the data distribution. These models only capture the local context, of up to a few thousands tokens. In this paper, we propose an extension of continuous cache models, which can scale to larger contexts. In particular, we use a large scale non-parametric memory component that stores all the hidden activations seen in the past. We leverage recent advances in approximate nearest neighbor search and quantization algorithms to store millions of representations while searching them efficiently. We conduct extensive experiments showing that our approach significantly improves the perplexity of pre-trained language models on new distributions, and can scale efficiently to much larger contexts than previously proposed local cache models.

## 1 Introduction

Language models are a core component of many natural language processing applications such as machine translation [3], speech recognition [2] or dialogue agents [50]. In recent years, deep learning has led to remarkable progress in this domain, reaching state of the art performance on many challenging benchmarks [31]. These models are known to be over-parametrized, and large quantities of data are needed for them to reach their full potential [12]. Consequently, the training time can be very long (up to weeks) even when vast computational resources are available [31]. Unfortunately, in many real-world scenarios, either such quantity of data is not available, or the distribution of the data changes too rapidly to permit very long training. A common strategy to circumvent these problems is to use a pre-trained model and slowly finetune it on the new source of data. Such adaptive strategy is also time-consuming for parametric models since the specificities of the new dataset must be slowly encoded in the parameters of the model. Additionally, such strategy is also prone to overfitting and dramatic forgetting of crucial information from the original dataset. These difficulties directly result from the nature of parametric models.

In contrast, non-parametric approaches do not require retraining and can efficiently incorporate new information without damaging the original model. This makes them particularly suitable for settings requiring rapid adaptation to a changing distribution or to novel examples. However, non-parametric models perform significantly worse than fully trained deep models [12]. In this work, we are interested in building a language model that combines the best of both non-parametric and parametric approaches: a deep language model to model most of the distribution and a non-parametric one to adapt it to the change of distribution.

This solution has been used in speech recognition under the name of cache models [36, 37]. Cache models exploit the unigram distribution of a recent context to improve the predictive ability of the model. Recently, Grave et al. [22] and Merity et al. [43] showed that this solution could be applied to neural networks. However, cache models depend on the local context. Hence, they can only adapt a parametric model to a local change in the distribution. These specificities limit their usefulness when

the context is unavailable (e.g., tweets) or is enormous (e.g., book reading). This work overcomes this limitation by introducing a fast non-parametric retrieval system into the hybrid approach. We demonstrate that this novel combination of a parametric neural language model with a non-parametric retrieval system can smoothly adapt to changes in the distribution while remaining as consistent as possible with the history of the data. Our approach is as a generalization of cache models which scales to millions of examples.

## 2 Related work

This section reviews different settings that require models to adapt to changes in the data distribution, like transfer learning or open set (continual) learning. We also discuss solutions specific to language models, and we briefly explain large-scale retrieval methods.

**Transfer Learning.** Transfer learning [10] is a well-established component of machine learning practitioners' toolbox. It exploits the commonalities between different tasks to improve the predictive performance of the models trained to solve them. Notable variants of transfer learning are multitask learning [10], domain adaptation [6], and curriculum learning [8]. Multitask learning jointly trains several models to promote sharing of statistical strength. Domain adaptation reuses existing information about a given problem (e.g., data or model) to solve a new task. Curriculum learning takes one step further by adapting an existing model across a (large) sequence of increasingly difficult tasks. Models developed for these settings have proven useful in practice. However, they are chiefly designed for supervised learning and do not scale to the size of the problem we consider in this work.

**Class-incremental and Open Set Learning.** These methods are concerned with problems where the set of targets is not known in advance but instead, increases over time. The main difficulty in this scenario lies in the deterioration of performance on previously seen classes when trying to accommodate new ones. Kuzborskij et al. [39] proposed to reduce the loss of accuracy when adding new classes by partly retraining the existing classifier. Muhlbaier et al. [47] introduced an ensemble model to deal with an increasingly large number of concepts. However, their approach relies on unrealistic assumptions on the data distribution. Zero-shot learning [41] can deal with new classes but often requires additional descriptive information about them [1]. Scheirer et al. [49] proposed a framework for open set recognition based on one-class SVMs.

**Adaptive language models.** Adaptive language models change their parameters according to the recent history. Therefore, they implement a form of domain adaptation. A popular approach adds a cache to the model and has shown early success in the context of speech recognition [36, 38, 37]. Jelinek et al. further extended this strategy [29] into a smoothed trigram language model, reporting a reduction in both perplexity and word error rates. Della Pietra et al.[15] adapt the cache to a general $n$-gram model such that it satisfies marginal constraints obtained from the current document. Closer to our work, Grave et al. [21] have shown that this strategy can improve modern language models like recurrent networks without retraining. However, their model assumes that the data distribution changes smoothly over time, by using a context window to improve the performance. Merity et al. [43] proposed a similar model, where the cache is jointly trained with the language model.

Other adaptive language models have been proposed in the past: Kneser and Steinbiss [35] and, Iyer and Ostendorf [26] dynamically adapt the parameters of their model to recent history using different weight interpolation schemes. Bellegarda [5] and Coccaro and Jurafsky [14] use latent semantic analysis to adapt their models to current context. Similarly, topic features have been used with either maximum entropy models [33] or recurrent networks [46, 53]. Finally, Lau et al. [42] propose to use pairs of distant of words to capture long-range dependencies.

**Large scale retrieval approaches.** The standard method for large-scale retrieval is to compress vectors and query them using a standard efficient algorithm. One of the most popular strategies is Locality-sensitive hashing (LSH) by Charikar [11], which uses random projections to approximate the cosine similarity between vectors by a function related to the Hamming distance between their corresponding binary codes. Several works have built on this initial binarization technique, such as spectral hashing [54], or Iterative Quantization (ITQ) [19]. Product Quantization (PQ) [28] approximates the distances between vectors by simultaneously learning the codes and the centroids, using

$k$-means. In the context of text, several works have shown that compression does not significantly reduce the performance of models [17, 24, 30].

# 3  Approach

In this section, we first briefly review language modeling and the use of recurrent networks for this task. We then describe our model, called *unbounded cache*, and explain how to scale it to large datasets with millions of words.

## 3.1  Language modeling

A language model evaluates the probability distribution of sequences of words. It is often framed as learning the conditional probability of words, given their history [4]. Let $V$ be the size of the vocabulary; each word is represented by a one-hot encoding vector $x$ in $\mathbb{R}^V = \mathcal{V}$, corresponding to its index in the dictionary. Using the chain rule, the probability assigned to a sequence of words $x_1, \ldots, x_T$ can be factorized as

$$p(x_1, ..., x_T) = \prod_{t=1}^{T} p(x_t \mid x_{t-1}, ..., x_1). \tag{1}$$

This conditional probability is traditionally approximated with non-parametric models based on counting statistics [20]. In particular, smoothed N-gram models [32, 34] have been the dominant type of models historically, achieving good performance in practice [44]. While the use of parametric models for language modeling is not new [48], their superiority has only been established with the recent emergence of neural networks [7, 45]. In particular, recurrent networks are now the standard approach, achieving state-of-the-art performances on several challenging benchmarks [31, 55].

## 3.2  Recurrent networks.

Recurrent networks are a special case of neural networks specifically designed for sequence modeling. At each time step, they maintain a hidden representation of the past and make a prediction accordingly. This representation is maintained by a continuous vector $h_t \in \mathbb{R}^d$ encoding the history $x_t, ..., x_1$. The probability of the next word is then simply parametrized using this hidden vector, i.e.,

$$p(w \mid x_t, ..., x_1) \propto \exp(h_t^\top o_w). \tag{2}$$

The *hidden* vector $h_t$ is computed by recursively applying an update rule:

$$h_t = \Phi\left(x_t, h_{t-1}\right), \tag{3}$$

where $\Phi$ is a function depending on the architecture of the network. Depending on $\Phi$, the hidden vectors may have a specific structure adapted to different sequence representation problems. Several architectures for recurrent networks have been proposed, such as the Elman network [16], the long short-term memory  (LSTM) [25] or the gated recurrent unit (GRU) [13]. For example, the Elman network [16] is defined by the following update rule

$$h_t = \sigma\left(Lx_t + Rh_{t-1}\right), \tag{4}$$

where $\sigma$ is a non-linearity such as the logistic or tanh functions, $L \in \mathbb{R}^{d \times V}$ is a word embedding matrix and $R \in \mathbb{R}^{d \times d}$ is the recurrent matrix. Empirical results have validated the effectiveness of the LSTM architecture to natural language modeling [31]. We refer the reader to [23] for details on this architecture. In the rest of this paper, we focus on this structure of recurrent networks.

Recurrent networks process a sentence one word at a time and update their weights by backpropagating the error of the prediction to a fixed window size of past time steps.  This training procedure is computationally expensive, and often requires a significant amount of data to achieve good performance. To circumvent the need of retraining such network for domain adaptation, we propose to add a non-parametric model that takes care of the fluctuation in the data distribution.

### 3.3 Unbounded cache

An unbounded cache adds a non-parametric and unconstrained memory to a neural network. Our approach is inspired by the cache model of Khun [36] and can be seen as an extension of Grave et al. [22] to an unbounded memory structure tailored to deal with out-of-vocabulary and rare words.

Similar to Grave et al. [22], we extend a recurrent neural network with a key-value memory component, storing the pairs $(h_i, w_{i+1})$ of hidden representation and corresponding word. This memory component also shares similarity with the parametric memory component of the pointer network introduced by Vinyals et al. [52] and extended by Merity et al. [43]. As opposed to these models and standard cache models, we do not restrict the cache component to recent history but store all previously observed words. Using the information stored in the cache component, we can obtain a probability distribution over the words observed up to time $t$ using the kernel density estimator:

$$p_{\text{cache}}(w_t \mid w_1, ...w_{t-1}) \propto \sum_{i=1}^{t-1} \mathbb{1}\{w = w_i\} K\left(\frac{\|h_t - h_i\|}{\theta}\right), \tag{5}$$

where $K$ is a kernel, such as Epanechnikov or Gaussian, and $\theta$ is a smoothing parameter. If $K$ is the Gaussian kernel ($K(x) = \exp(-x^2/2)$) and the hidden representations are normalized, this is equivalent to the continuous cache model.

As the memory grows with the amount of data seen by the model, this probability distribution becomes impossible to compute. Millions of words and their multiple associated context representations are stored, and exact exhaustive matching is prohibitive. Instead, we use the approximate $k$-nearest neighbors algorithm that is described below in Sec. 3.4 to estimate this probability distribution:

$$p_{\text{cache}}(w_t \mid w_1, ...w_{t-1}) \propto \sum_{i \in \mathcal{N}(h_t)} \mathbb{1}\{w = w_i\} K\left(\frac{\|h_t - h_i\|}{\theta(h_t)}\right), \tag{6}$$

where $\mathcal{N}(h_t)$ is the set of nearest neighbors and $\theta(h_t)$ is the Euclidean distance from $h_t$ to its $k$-th nearest neighbor. This estimator is known as variable kernel density estimation [51]. It should be noted that if the kernel $K$ is equal to zero outside of $[-1, 1]$, taking the sum over the $k$ nearest neighbors is equivalent to taking the sum over the full data.

The distribution obtained using the estimator defined in Eq. 6 assigns non-zero probability to at most $k$ words, where $k$ is the number of nearest neighbors used. In order to have non-zero probability everywhere (and avoid getting infinite perplexity), we propose to linearly interpolate this distribution with the one from the model:

$$p(w_t \mid w_1, ...w_{t-1}) = (1 - \lambda)p_{\text{model}}(w_t \mid w_1, ...w_{t-1}) + \lambda p_{\text{cache}}(w_t \mid w_1, ...w_{t-1}).$$

### 3.4 Fast large scale retrieval

Fast computation of the probability of a rare word is crucial to make the cache grow to millions of potential words. Their representation also needs to be stored with relatively low memory usage. In this section, we briefly describe a scalable retrieval method introduced by Jegou et al. [27]. Their approach called Inverted File System Product Quantization (IVFPQ) combines two methods, an inverted file system [56] and a quantization method, called Product quantization (PQ) [28]. Combining these two components offers a good compromise between a fast retrieval of approximate nearest neighbors and a low memory footprint.

**Inverted file system.** Inverted file systems [56] are a core component of standard large-scale text retrieval systems, like search engines. When a query $x$ is compared to a set $\mathcal{Y}$ of potential elements, an inverted file avoids an exhaustive search by providing a subset of possible matching candidates. In the context of continuous vectors, this subset is obtained by measuring some distance between the query and predefined vector representations of the set. More precisely, these candidates are selected through "coarse matching" by clustering all the elements in $\mathcal{Y}$ in $c$ groups using $k$-means. The centroids are used as the vector representations. Each element of the set $\mathcal{Y}$ is associated with one centroid in an inverted table. The query $x$ is then compared to each centroid and a subset of them is selected according to their distance to the query. All the elements of $\mathcal{Y}$ associated with these centroids are then compared to the query $x$. Typically, we take $c$ centroids and keep the $c_c$ closest centroids to a query.

This procedure is quite efficient but very memory consuming, as each vector in the set $\mathcal{Y}$ must be stored. This can be drastically reduced by quantizing the vectors. Product Quantization (PQ) is a popular quantization method that has shown competitive performance on many retrieval benchmarks [28]. Following Jegou et al. [28], we do not directly quantize the vector $y$ but its residual $r$, i.e., the difference between the vector and its associated centroids.

**Product Quantization.** Product quantization is a data-driven compression algorithm with no overhead during search [28]. While PQ has been designed for image feature compression, Joulin et al. [30] have demonstrated its effectiveness for text too. PQ compresses real-valued vector by approximating them with the closest vector in a pre-defined structured set of centroids, called a codebook. This codebook is obtained by splitting each residual vector $r$ into $k$ subvectors $r^i$, each of dimension $d/k$, and running a $k$-means algorithm with $s$ centroids on each resulting subspace. The resulting codebook contains $c^s$ elements which is too large to be enumerated, and is instead implicitly defined by its structure: a $d$-dimensional vector $x \in \mathbb{R}^d$ is approximated as

$$\hat{x} = \sum_{i=1}^{k} q_i(x), \tag{7}$$

where $q_i(x)$ is the closest centroid to subvector $x^i$. For each subspace, there are $s = 2^b$ centroids, where $b$ is the number of bits required to store the quantization index of the sub-quantizer. Note that in PQ, the subspaces are aligned with the natural axis and improvements where made by Ge et al. [18] to align the subspaces to principal axes in the data. The reconstructed vector can take $2^{kb}$ distinct reproduction values and is stored in $kb$ bits.

PQ estimates the inner product in the compressed domain as

$$x^\top y \approx \hat{x}^\top y = \sum_{i=1}^{k} q_i(x^i)^\top y^i. \tag{8}$$

In practice, the vector estimate $\hat{x}$ is trivially reconstructed from the codes, (i.e., from the quantization indexes) by concatenating these centroids. PQ uses two parameters, namely the number of sub-quantizers $k$ and the number of bits $b$ per quantization index.

## 4    Experiments

In this section, we present evaluations of our unbounded cache model on different language modeling tasks. We first briefly describe our experimental setting and the datasets we used, before presenting the results.

### 4.1    Experimental setting

One of the motivations of our model is to be able to adapt to changing data distribution. In particular, we want to incorporate new words in the vocabulary, as they appear in the test data. We thus consider a setting where we do not replace any words by the <unk> token, and where the test set contains out-of-vocabulary words (OOV) which were absent at train time. Since we use the perplexity as the evaluation metric, we need to avoid probabilities equal to zero in the output of our models (which would result in infinite perplexity). Thus, we always interpolate the probability distributions of the various models with the uniform distribution over the full vocabulary:

$$p_{\text{uniform}}(w_t) = \frac{1}{|\text{vocabulary}|}.$$

This is a standard technique, which was previously used to compare language models trained on datasets with different vocabularies [9].

**Baselines** We compare our unbounded cache model with the static model interpolated with uniform distribution, as well as the static model interpolated with the unigram probability distribution observed up to time $t$. Our proposal is a direct extension of the local cache model [22]. Therefore, we also compare to it to highlight the settings where an unbounded cache model is preferable to a local one.

| model | Size | OoV rate (%) |
|---|---|---|
| News 2008 | 219,796 | 2.3% |
| News 2009 | 218,628 | 2.4% |
| News 2010 | 205,859 | 2.4% |
| News 2011 | 209,187 | 2.5% |
| Commentary | 144,197 | 4.2% |
| Web | 321,072 | 5.9% |
| Wiki | 191,554 | 5.5% |
| Books | 174,037 | 3.7% |

Table 1: Vocabulary size and out-of-vocabulary rate for various test sets (for a model trained on News 2007).

## 4.2 Implementation details

We train recurrent neural networks with 256 LSTM hidden units, using the Adagrad algorithm with a learning rate of $0.2$ and $10$ epochs. We compute the gradients using backpropagation through time (BPTT) over 20 timesteps. Because of the large vocabulary sizes, we use the adaptative softmax [21]. We use the IVFPQ implementation from the FAISS open source library.[1] We use $4,096$ centroids and $8$ probes for the inverted file. Unless said otherwise, we query the $1,024$ nearest neighbors.

## 4.3 Datasets

Most commonly used benchmarks for evaluating language models propose to replace rare words by the <unk> token. On the contrary, we are interested in open vocabulary settings, and therefore decided to use datasets without <unk>. We performed experiments on data from the five following domains:

- **News Crawl**[2] is a dataset made of news articles, collected from various online publications. There is one subset of the data for each year, from 2007 to 2011. This dataset will allow testing the unbounded cache models on data whose distribution slowly changes over time. The dataset is shuffled at the sentence level. In the following, we refer to this dataset as `news 2007-2011`.

- **News Commentary** consists of political and economic commentaries from the website `https://www.project-syndicate.org/`. This dataset is publicly available from the Statistical Machine Translation workshop website. In the following, we refer to this dataset as `commentary`.

- **Common Crawl** is a text dataset collected from diverse web sources. The dataset is shuffled at the sentence level. In the following, we refer to this dataset as `web`.

- **WikiText**[3] is a dataset derived from high quality English Wikipedia articles, introduced by Merity et al. [43]. Since we do not to replace any tokens by <unk>, we use the raw version. In the following, we refer to this dataset as `wiki`.

- **The book Corpus** This is a dataset of 3,036 English books, collected from the Project Gutenberg[4] [40]. We use a subset of the books, which have a length around 100,000 tokens. In the following we refer to this dataset as `books`.

All these datasets are publicly available. Unless stated otherwise, we use 2 million tokens for training the static models and 10 million tokens for evaluation. All datasets are lowercased and tokenized using the europarl dataset tools.[5]

| | | Test set | | | | |
|---|---:|---:|---:|---:|---:|
| model | 2007 | 2008 | 2009 | 2010 | 2011 |
| static | 220.9 | 237.6 | 256.2 | 259.7 | 268.8 |
| static + unigram | 220.3 | 235.9 | 252.6 | 256.1 | 264.3 |
| static + local cache | 218.9 | 234.5 | 250.5 | 256.2 | 265.2 |
| static + unbounded cache | 166.5 | 191.4 | 202.6 | 204.8 | 214.3 |

Table 2: Static model trained on `news` 2007 and tested on `news` 2007-2011.

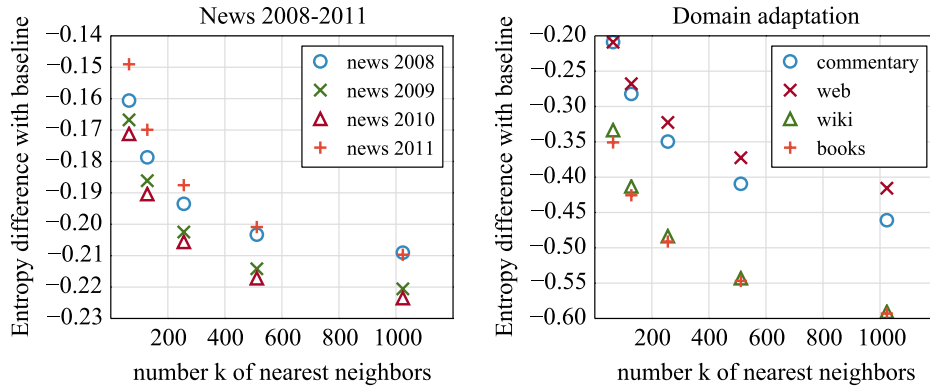

Figure 1: Performance of our model, as a function of the number $k$ of nearest neighbors, used to estimate the probability of words in the unbounded cache. We report the entropy difference with the static+unigram baseline.

| | | | Test domain | | | |
|---|---|---:|---:|---:|---:|---:|
| Train domain | model | News | Commentary | Web | Wiki | Books |
| | static | - | 342.7 | 689.3 | 1003.2 | 687.1 |
| News | static + unigram | - | 303.5 | 581.1 | 609.4 | 349.1 |
| | static + local cache | - | 288.5 | 593.4 | **316.5** | 240.3 |
| | static + unbounded cache | - | **191.1** | **383.4** | 337.4 | **237.2** |
| | static | 624.1 | 484.0 | - | 805.3 | 784.3 |
| Web | static + unigram | 519.2 | 395.6 | - | 605.3 | 352.4 |
| | static + local cache | 531.4 | 391.3 | - | **321.5** | 235.8 |
| | static + unbounded cache | **306.3** | **234.9** | - | 340.2 | **223.6** |
| | static | 638.1 | 626.3 | 901.0 | - | 654.6 |
| Wiki | static + unigram | 537.9 | 462.2 | 688.5 | - | 346.9 |
| | static + local cache | 532.8 | 436.7 | 694.3 | - | 228.8 |
| | static + unbounded cache | **318.7** | **255.3** | **456.1** | - | **223.8** |

Table 3: Static model trained on `news` 2007 and tested on data from other domains.

| Dataset | Static model | Local cache | Unbounded cache |
|---|---:|---:|---:|
| `News 2008` | 82 | 664 | 433 |
| `Commentary` | 78 | 613 | 494 |
| `Web` | 85 | 668 | 502 |
| `Wiki` | 87 | 637 | 540 |
| `Books` | 81 | 626 | 562 |

Table 4: Computational time (in seconds) to process 10M tokens from different test sets for the static language model, the local cache (size 10,000) and the unbounded cache.

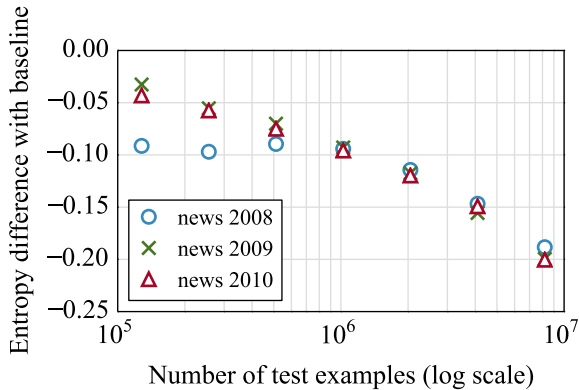

Figure 2: Performance of the un-bounded cache model, as a function of the number of test examples. We report the entropy difference with the static+unigram baseline. We observe that, as the number of test examples increases (and thus, the information stored in the cache), the performance of the unbounded cache increases.

## 4.4 Results

We demonstrate the effectiveness of using an unbounded cache to complement a language model as advocated in the previous sections model by performing two types of experiments representing a *near domain* and *far domain* adaptation scenarios. In both experiments, we compare the unigram static model, the unigram extension, and the unbounded cache model.

**Local vs. Unbounded Cache**   We first study the impact of using an unbounded cache instead of a local one. To that end, we compare the performance of the two models when trained and tested on different combinations of the previously described datasets. These datasets can be categorized into two groups according to their properties and the results obtained by the various models we use.

On the one hand, the Wiki and Books datasets are not shuffled. Hence, the recent history (up to a few thousands words) contains a wealth of information that can be used by a local cache to reduce the perplexity of a static model. Indeed, the local cache model achieves respectively 316.5 and 240.3 on the Wiki and Books datasets when trained on the News dataset. This corresponds to about $3\times$ reduction in perplexity on both datasets in comparison to the static model. A similar trend holds when the training data is either Web or Wiki dataset. Surprisingly, the unbounded cache model performs similarly to the cache model despite using orders of magnitude broader context. A static model trained on News and augmented with an unbounded cache achieves respectively 337.4 and 237.2 of perplexity. It is also worth noting that our approach is more efficient than the local cache, while storing a much larger number of elements. Thanks to the use of fast nearest neighbor algorithm, it takes 502 seconds to process 10M tokens from the test set when using the unbounded cache. Comparatively, it takes 668 seconds for a local cache model of size $10,000$ to perform a similar task. The timing experiments, reported in Table 4.3, show a similar trend.

On the other hand, the Commentary and Web datasets are shuffled. Therefore, a local cache can hardly capture the relevant statistics to significantly improve upon the static model interpolated with the unigram distribution. Indeed, the perplexity of a local cache model on these datasets when the static model is trained on the News dataset is respectively 288.5 and 593.4. In comparison, the unbounded cache model achieves on the same datasets respectively a perplexity of 191.1 and 383.4. That is an average improvement of about $50\%$ over the local cache in both cases (see Table 3).

**Near domain adaptation.**   We study the benefit of using an unbounded cache model when the test domain is only slightly different from the source domain. We train the static model on `news 2007` and test on the corpus `news 2008 to news 2011`. All the results are reported in Table 1.

We first observe that the unbounded cache brings a $24.6\%$ improvement relative to the static model on the in-domain `news 2007` corpus by bringing the perplexity from 220.9 down to 166.5. In comparison, neither using the unigram information nor using a local cache lead to significant improvement. This result underlines two phenomena. First, the simple distributional information captured by the unigram or the local cache is already captured by the static model. Second, the unbounded cache enhances the discrimination capabilities of the static model by capturing useful non-linearities thanks to the combination of the nearest neighbor and the representation extracted from

the static model. Interestingly, these observations remain consistent when we consider evaluations on the test sets `news 2008-2011`. Indeed, the average improvement of unbounded cache relatively to the static model on the corpus `news 2008-2011` is $20.44\%$ while the relative improvement of the unigram cache is only $1.3\%$. Similarly to the in-domain experiment, the unigram brings little useful information to the static model mainly because the source (`news 2007`) and the target distributions (`news 2008-2011`) are very close. In contrast, the unbounded cache still complements the static model with valuable non-linear information of the target distributions.

**Far domain adaptation.** Our second set of experiments is concerned with testing on different domains from the one the static model is trained on. We use the `News`, `Web` and `Wiki` datasets as source domains, and all five domains as target. The results are reported in Table 3.

First, we observe that the unigram, the local and the unbounded cache significantly help the static model in all the far domain adaptation experiments. For example, when adapting the static model from the `News` domain to the `Commentary` and `Wiki` domains, the unigram reduces the perplexity of the static model by 39.2 and 393.8 in absolute value respectively. The unbounded cache significantly improves upon the static model and the unigram on all the far domain adaptation experiment. The smallest relative improvement compared to the static model and the unigram is achieved when adapting from `News` to `Web` and is $79.7\%$ and $51.6\%$ respectively. The more the target domain is different from the source one, the more interesting is the use of an unbounded cache mode. Indeed, when adapting to the `Books` domain (which is the most different from the other domains) the average improvement given by the unbounded cache relatively to the static model is $69.7\%$.

**Number of nearest neighbors.** Figure 1 shows the performance of our model with the number of nearest neighbors per query. As observed previously by Grave et al [22], the performance of a language model improves with the size of the context used in the cache. This context is, in some sense, a constrained version of our set of retained nearest neighbors. Interestingly, we observe the same phenomenon despite forming the set of possible predictions over a much broader set of potential candidates than the immediate local context. Since IFVPQ has a linear complexity with the number of nearest neighbors, setting the number of nearest neighbors to a thousand offers a good trade-off between speed and accuracy.

**Size of the cache.** Figure 2 shows the gap between the performance of static language model with and without the cache as the size of the test set increases. Despite having a much more significant set of candidates to look from, our algorithm continues to select relevant information. As the test set is explored, better representations for rare words are stored, explaining this constant improvement.

## 5   Conclusion

In this paper, we introduce an extension to recurrent networks for language modeling, which stores past hidden activations and associated target words. This information can then be used to obtain a probability distribution over the previous words, allowing the language models to adapt to the current distribution of the data dynamically. We propose to scale this simple mechanism to large amounts of data (millions of examples) by using fast approximate nearest neighbor search. We demonstrated on several datasets that our unbounded cache is an efficient method to adapt a recurrent neural network to new domains dynamically, and can scale to millions of examples.

**Acknowledgements**

We thank the anonymous reviewers for their insightful comments.

## Footnotes

[1] https://github.com/facebookresearch/faiss

[2] http://www.statmt.org/wmt14/translation-task.html

[3] https://metamind.io/research/the-wikitext-long-term-dependency-language-modeling-dataset/

[4] http://www.gutenberg.org/

[5] http://statmt.org/europarl/v7/tools.tgz

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
