[Reviews · NeurIPS 2017]

Reviewer 1



This paper proposes a non-parametric method to cache previously seen contexts for language modelling. The basic idea is that at each point, the k nearest-neighbour states from previously seen contexts are retrieved, and a kernel density estimation method applied to generate a probability distribution over an open vocabulary. Thus, the cache model is unbounded, unlike methods such as pointer networks or continuous caches. Results demonstrate the performance of this method on language modelling with time and topic drift, over standard RNN language models. This was a well-written paper, and the unbounded cache idea is intuitively appealing. I think that the proposed method could become useful for many tasks besides language modelling. I would have liked to see a comparison of this method against parametric or local cache methods such as the pointer-generator network. Also, how much slower is the proposed model at inference time? Querying 1024 nearest neighbours in order to estimate p_{cache} looks like it may be expensive.

Reviewer 2



This paper proposes a modification of recurrent networks (Elman network, LSTM or gated recurrent unit) for language modelling that is able to adapt to changes in the data distribution, by introducing an unbounded cache defined to deal with unseen words during training or uncommon words. Although there are other similar works in the literature (see [21],[50],[41]), this one allows to store all the previously seen words, instead of just the most recent ones by estimating the probability distribution of the words seen using a kernel density estimator (and an approximate knn to scale the process). From a theoretical point of view, the paper is interesting and addresses an issue that is present in many NLP applications. In the result sections, the authors show that using the cache is important to obtain good results. During the reviewing process they have added substantial empirical validation to the paper, showing that it improves on the state of the art. Opinion on the paper: + the paper is very well written (although the authors should pass an english corrector) + the topic is of great importance for every NLP practitioner - the paper lacks proper numerical evaluation (although in the rebuttal process the authors provided a wide range of numerical results, they cannot be included in the final paper, thus not taken into account in the evaluation)

Reviewer 3



This paper discusses an extensions to the recently proposed continuous cache models by Grave et al. The authors propose a continuous cache model that is unbounded, hence can take into account events that happened an indefinitely long time ago. While interesting, the paper fails to provide good experimental evidence of its merits. Its main statement is that this model is better than Grave et al., but then does not compare with it. It only seems to compare with cache models from the nineties (Kuhn et al.), although that is not even clear as they spend only one line (line 206) discussing the models they compare with. "the static model interpolated with the unigram probability distribution observed up to time t" does sound like Kuhn et al. and is definitely not Grave et al. The authors also mention the importance of large vocabularies, yet fail to specify the vocabulary size for any of their experiments. I also don't understand why all the datasets were lowercased if large vocabularies are the target? This (and the above) is a real pity, because sections 1-3 looked very promising. We advise the authors to spend some more care on the experiments section to make this paper publishable. Minor comments: * line 5: stores -> store * line 13: twice "be" * line 30: speach -> speech * line 31: "THE model" * line 59: "assumptions ABOUT" * line 72: no comma after "and" * line 79: algorithm -> algorithmS * line 79: approach -> approachES * line 97: non-parameteric -> non-parametric * line 99: "THE nineties" * line 110: "aN update rule" * line 127: Khun -> Kuhn * line 197: motivationS * line 197: "adapt to A changing distribution" * line 211: "time steps" * line 211: adaptative -> adaptive * table 2: the caption mentions that the model is trained on news 2007, but in fact this varies throughout the table? * line 216: interest -> interested * line 235: millions -> million * line 267: experimentS * line 269: where do these percentages come from? they seem wrong... * line 281: "THE static model" * line 283: Set